# Ensembles of Random SHAPs

**Lev Utkin \*,† and Andrei Konstantinov †**

Higher School of Artificial Intelligence, Institute of Computer Science and Technology, Peter the Great St. Petersburg Polytechnic University, Polytechnicheskaya, 29, 195251 St. Petersburg, Russia

\* Correspondence: utkin_lv@spbstu.ru

† These authors contributed equally to this work.

**Abstract:** The ensemble-based modifications of the well-known SHapley Additive exPlanations (SHAP) method for the local explanation of a black-box model are proposed. The modifications aim to simplify the SHAP which is computationally expensive when there is a large number of features. The main idea behind the proposed modifications is to approximate the SHAP by an ensemble of SHAPs with a smaller number of features. According to the first modification, called the ER-SHAP, several features are randomly selected many times from the feature set, and the Shapley values for the features are computed by means of "small" SHAPs. The explanation results are averaged to obtain the final Shapley values. According to the second modification, called the ERW-SHAP, several points are generated around the explained instance for diversity purposes, and the results of their explanation are combined with weights depending on the distances between the points and the explained instance. The third modification, called the ER-SHAP-RF, uses the random forest for a preliminary explanation of the instances and determines a feature probability distribution which is applied to the selection of the features in the ensemble-based procedure of the ER-SHAP. Many numerical experiments illustrating the proposed modifications demonstrate their efficiency and properties for a local explanation.

**Keywords:** explanation model; XAI; SHAP; random forest; ensemble model

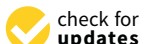

## 1. Introduction

Machine learning models and algorithms have shown increasing importance and success in many domains. Despite the success, there are obstacles for applying machine learning algorithms, especially in areas of risk, for example, in medicine, reliability maintenance, autonomous vehicle systems, and security applications. One of the obstacles is that many machine learning models have sophisticated architectures and, therefore, they are viewed as black boxes. As a result, models have a limited interpretability, and a user of the corresponding model cannot understand and explain the predictions and decisions provided by the model. Another obstacle is that a single testing instance has to be explained in many cases, i.e., a user needs to understand only a single prediction, for example, a diagnosis of a patient stated by a model. In order to overcome these obstacles, additional interpretable models should be developed that could help to answer the question, which features of an analyzed instance lead to the black-box survival model prediction. In other words, these models should select the most important features which impact the black-box model prediction. It should be noted that some models, including linear regression, logistic regression, and decision trees, are intrinsically explainable due to their peculiarities. At the same time, most machine learning models, especially deep learning models, are black boxes and cannot be directly explained. An explanation of these models and their predictions motivated developing a lot of methods and models which try to explain the predictions of the deep classification and regression algorithms. There are several detailed survey papers providing a deep dive into the variety of interpretation methods and models [1–8],

which show the increasing importance of the interpretation methods and a growing interest in them.

The interpretation of the black-model local prediction aims to select features which significantly impact on this prediction, i.e., by using the interpretation model, we try to determine *which features of an analyzed instance lead the obtained black-box model prediction*. There are two groups of interpretation methods. The first one consists of the so-called local methods. They try to interpret a black-box model locally around a test instance. The second group contains global methods which derive interpretations on the whole dataset or its part. The present paper focuses on the first group of local interpretation methods, though the proposed approach can be simply extended to the global interpretation.

Two very popular post hoc approaches to interpretation can be selected among many others. The first one is LIME (Local Interpretable Model-Agnostic Explanation) [9], which is based on building an approximating linear model around the instance to be explained. This follows from the intuition that the explanation may be derived locally from many instances generated in the neighborhood of the explained instance with weights defined by their distances from the explained instance. The coefficients of the linear model are interpreted as the feature's importance. The linear regression for solving the regression problem or the logistic regression for solving the classification problem allow us to construct the corresponding linear models. LIME has many advantages. It successfully interprets models dealing with tabular data, text, and images. However, there are some shortcomings of LIME. The first one is that LIME is not robust. This means that it may provide very different explanations for two nearby data points. The definition of neighborhoods is also very vague. Moreover, LIME may provide an incorrect explanation when there is a small difference between the training and testing data. LIME is also sensitive to the parameters of the explanation model, for example, to the weights of the generated instances, to the number of the generated instances, etc.

The second approach consists of the well-known SHAP (SHapley Additive exPlanations) method [10,11] and its modifications. The method is inspired by game-theoretic Shapley values [12] which can be interpreted as average expected marginal contributions over all possible subsets (coalitions) of features to the black-box model prediction. The SHAP has many advantages, for example, it can be used for local and global explanations in contrast to LIME, but there are also two important shortcomings. The first one is a question as to how to add or remove features in order to implement their subsets as inputs for the black-box model. There are many approaches to removing features, exhaustively described by [13], but the SHAP may be too sensitive to each of them, and there are no strong justifications for their use. Nevertheless, the SHAP can be regarded as the most promising and efficient explanation method.

The second shortcoming is that the SHAP is computationally expensive when there is a large number of features due to considering all possible coalitions whose number is $2^m$, where $m$ is the number of features. Therefore, the computational time grows exponentially. Several simplifications and approximations have been proposed in order to overcome this difficulty. Some of them are presented by [11,14,15]. One of the simplifications is based on using the ordered permutations of the feature indices and the probability distributions of the features [14]. Another approximation is the quasi-random and adaptive sampling which includes two improvements [15]. The first one is based on exploiting the Monte Carlo integration. The second improvement is based on the optimal number of the perturbations of every feature in accordance with its variance to minimize the overall approximation error. Ref. [15] also proposed to average the local contributions of the values of each feature across all instances. Another interesting approach to simplify the SHAP is the Kernel SHAP [10] which can be regarded as a computationally efficient approximation to the Shapley values in higher dimensions. In order to relax the assumption of the feature independence accepted in the Kernel SHAP, ref. [16] extended the Kernel SHAP method to handle dependent features. Ref. [17] proposed the polynomial-time approximation of the Shapley values, called the Deep Approximate Shapley Propagation method.

In spite of the many approaches to simplify the SHAP, it is difficult to expect a significant simplification from the above modifications of the SHAP. Therefore, a new approach is proposed for simplifying the SHAP method and for reducing the computational expenses for calculating the Shapley values. A key idea behind the proposed approach is to apply a modification of the random subspace method [18] and to consider an ensemble of random SHAPs, called the ensemble of random SHAPs (ER-SHAP). The approach is very similar to the random forests, when an ensemble of randomly built decision trees is used to obtain some average classification or regression measures. Random SHAPs are constructed by a random selection of $t$ features with indices $J_k = (i_1, \ldots, i_t)$ from the instance for an explanation, and the obtained subset of the instance features is analyzed by the SHAP as a separate instance. Repeating this procedure $N$ times, we obtain a set $\mathcal{S} = \{S_1, \ldots, S_N\}$ of the Shapley values corresponding to the input subsets of the features, where the $k$-th subset is $S_k = \{\phi_i : i \in J_k\}$. By applying some combination rule for combining subsets $S_k$ from $\mathcal{S}$, we obtain the final Shapley values.

The above general approach considering an ensemble of random SHAPs has several extensions which form the corresponding methods and algorithms. First of all, we can generate points around the analyzed instance and construct $S_k$ for the $k$-th generated point. In this case, every point is assigned by a weight, depending on the distance from the analyzed point. As a result, we can combine the subsets $S_k$ of the Shapley values with weights which are defined as a function of the distance from the analyzed point. This modification is called the ensemble of random weighted SHAPs (ERW-SHAP).

Another extension or modification is to select features in accordance with a probability distribution to obtain instances consisting of features with indices from the set $J_k$. Let us define the discrete probability distribution over the set of all indices. It can be produced, for example, by using the random forest [19] which plays the role of a feature selection model. At that, the random forest is constructed by using a set of points (instances) locally generated around the explained point. Every decision tree is built by using a single point from the set of the generated points. This modification is called the ensemble of random SHAPs generated by the random forest (ER-SHAP-RF).

In sum, the contribution of this paper can be formulated as follows:

1.  A new approach to implementing an ensemble-based SHAP with random subsets of features of the explained instance is proposed.
2.  Several combination schemes are studied for aggregating the subsets of the important features obtained by using random SHAPs.
3.  The approach is extended by generating random points in the local area around a test instance and computing the subsets of the important features separately for every point. Some kind of diversity is implemented with this extension.
4.  Another extension is to use a probability distribution for the random selection of the features defined by the means of the random forest constructed by using the generated points in the local area around a test instance. The preliminary feature selection can be regarded as a pre-training procedure.

A lot of numerical experiments with an algorithm implementing the proposed method on synthetic and real datasets demonstrate its efficiency and the properties for the local and global interpretation.

This paper is organized as follows. The related work is in Section 2. The Shapley values and the SHAP method as a powerful tool for local and global explanations are introduced in Section 3. A detailed description of the proposed modifications of the SHAP, including the ER-SHAP, ERW-SHAP, and ER-SHAP-RF, is provided in Section 4. The numerical experiments with synthetic data and real data using the local interpretation by means of the proposed models and their comparison with the standard SHAP method are given in Section 5. The concluding remarks can be found in Section 6.

## 2. Related Work

The increasing importance of machine learning models and algorithms leads to the development of new explanation methods taking into account the various peculiarities of applied problems. Among the various approaches, we consider the local interpretation models which aim to explain a specific decision or a prediction obtained for a single instance. The local interpretation is especially important in medicine where a diagnosis of a patient has to be confirmed. As a result, many models of the local interpretation have been proposed. The success and simplicity of the LIME interpretation method resulted in the development of several of its modifications, for example, ALIME [20], Anchor LIME [21], LIME-Aleph [22], GraphLIME [23], SurvLIME [24], etc. A comprehensive analysis of LIME, including the study of its applicability to different data types, for example, text and image data, was provided by [25]. The same analysis for tabular data was proposed by the same authors [26]. An image version of LIME with its deep theoretical study was presented by [27]. An interesting information-theoretic justification of interpretation methods on the basis of the concept of the explainable empirical risk minimization was proposed by [28].

In order to relax the linearity condition for the local interpretation models like LIME and to adequately approximate a black-box model, several interpretation methods based on using Generalized Additive Models [29] were proposed [30–33]. Another interesting class of models based on using a linear combination of neural networks, such that a single feature is fed to each network, was proposed by [34]. The impact of every feature on the prediction in these models is determined by its corresponding shape function obtained by each neural network. Following the ideas behind these interpretation models, [35] proposed a similar model. In contrast to the method proposed by [34], an ensemble of gradient boosting machines was used in [35] instead of neural networks in order to simplify the explanation model training process.

Another explanation method was the SHAP [10,11], which takes a game-theoretic approach for optimizing a regression loss function based on the Shapley values. General questions of the computational efficiency of the SHAP were investigated by [36]. Ref. [37] proposed the generalized SHAP method which allows us to compute the feature importance of any function of a model's output. Ref. [38] presented an approach to applying the SHAP to ensemble models. The problem of explaining the predictions of graph neural networks by using the SHAP was considered by [39]. Ref. [40] introduced the so-called off- and on-manifold Shapley values for high-dimensional multi-type data. The application of the SHAP to the explanation of recurrent neural networks was studied in [41]. Ref. [42] presented a new approach to explaining fairness in machine learning, based on the Shapley value paradigm. Ref. [43] studied how to explain the anomalies detected by autoencoders using the SHAP. The problem of explaining the anomalies detected by a PCA was also considered by [44]. Ref. [45] proposed the X-SHAP which extends one of the approximations of the SHAP called the Kernel SHAP [10]. The SHAP was also applied to the problems of explaining individual predictions when features are dependent [16] or when features are mixed [46]. The SHAP was used in real applications to explain the predictions of the black-box models, for example, it was used to rank the failure modes of reinforced concrete columns and to explain why a machine learning model predicts a specific failure mode for a given sample [47]. It was also used in chemoinformatics and medicinal chemistry [48]. An interesting application of the SHAP in the desirable interpretation of the machine learning-based model results for identifying m7G sites in the gene expression analysis was proposed by [49]. The basic problems of the SHAP were also analyzed by [50].

Many other interpretation methods, their analyses, and critical reviews can also be found in survey papers [1–3,6,51–55].

## 3. Shapley Values and the Explanation Model

One of the most powerful approaches to explaining predictions of the black-box machine learning models is the approach based on using the Shapley values [12] as a key concept in coalitional games. According to the concept, the total gain of a game is

distributed to players such that desirable properties, including efficiency, symmetry, and linearity, are fulfilled. In the framework of the machine learning, the gain can be viewed as the machine learning model prediction or the model output, and a player is a feature of input data. Hence, contributions of features to the model prediction can be estimated by Shapley values. The $i$-th feature importance is defined by the Shapley value

$$\phi_i(f) = \phi_i = \sum_{S \subseteq N \setminus \{i\}} B(S, N)[f(S \cup \{i\}) - f(S)], \tag{1}$$

where $f(S)$ is the characteristic function in terms of coalitional games or the black-box model prediction under condition that a subset $S$ of features are used as the corresponding input in terms of machine learning; $N$ is the set of all features; $B(S, N)$ is defined as

$$B(S, N) = \frac{|S|!(|N| - |S| - 1)!}{|N|!}. \tag{2}$$

It can be seen from the above expression that the Shapley value $\phi_i$ can be regarded as the average contribution of the $i$-th feature across all possible permutations of the feature set.

The Shapley value has the following important properties:

**Efficiency**. The total gain is distributed as $\sum_{k=0}^{m} \phi_k = f(\mathbf{x})$.

**Symmetry**. If two players with numbers $i$ and $j$ make equal contributions, i.e., $f(S \cup \{i\}) = f(S \cup \{j\})$ for all subsets $S$ which contain neither $i$ nor $j$, then $\phi_i = \phi_j$.

**Dummy**. If a player makes zero contributions, i.e., $f(S \cup \{j\}) = f(S)$ for a player $j$ and all $S \subseteq N \setminus \{j\}$, then $\phi_j = 0$.

**Linearity**. A linear combination of multiple games $f_1, \ldots, f_n$, represented as $f(S) = \sum_{k=1}^{n} c_k f_k(S)$, has gains derived from $f$: $\phi_i(f) = \sum_{k=1}^{m} c_k \phi_i(f_k)$ for every $i$.

Let us consider a machine learning problem. Suppose that there is a dataset $\{(\mathbf{x}_1, y_1), \ldots, (\mathbf{x}_n, y_n)\}$ of $n$ points $(\mathbf{x}_i, y_i)$, where $\mathbf{x}_i \in \mathcal{X} \subset \mathbb{R}^m$ is a feature vector consisting of $m$ features, $y_i$ is the observed output for the feature vector $\mathbf{x}_i$ such that $y_i \in \mathbb{R}$ in the regression problem and $y_i \in \{1, 2, \ldots, T\}$ in the classification problem with $T$ classes. If a task is to interpret or to explain the prediction from the model $f(\mathbf{x}^*)$ at a local feature vector $\mathbf{x}^*$, then the prediction $f(\mathbf{x}^*)$ can be represented by using Shapley values as follows [10,11]:

$$f(\mathbf{x}^*) = \phi_0 + \sum_{j=0}^{m} \phi_j^*, \tag{3}$$

where $\phi_0 = \mathbb{E}[f(\mathbf{x})]$, $\phi_j^*$ is the value $\phi_j$ for the prediction $\mathbf{x} = \mathbf{x}^*$.

The above implies that the Shapley values explain the difference between the prediction $f(\mathbf{x}^*)$ and the global average prediction.

One of the crucial questions for implementing the SHAP method is how to remove features from subset $N \setminus S$, i.e., how to fill input features from subset $N \setminus S$ in order to obtain predictions $f(S)$ of the black-box model. A detailed description of the various ways for removing features is presented by [13]. One of the ways is simply by setting the removed features to zero [56,57] or by setting them to user-defined default values [9]. According to this way, features are often replaced with their mean values. Another way removes a feature by replacing them with a sample from a conditional generative model [58]. In the LIME method for tabular data, features are replaced with independent draws from specific distributions [13] such that each distribution depends on the original feature values. These are only a part of all the ways of removing features.

## 4. Modifications of SHAP

### 4.1. Ensemble of Random SHAPs

In spite of the many approaches to simplify SHAP, it is difficult to expect a significant simplification from the above modifications of SHAP. Therefore, a new approach is

proposed for simplifying the SHAP method and for reducing computational expenses for calculating the Shapley values. A key idea behind the proposed approach is to apply a modification of the random subspace method [18] and to consider an ensemble of random SHAPs. The approach is very similar to the random forests when an ensemble of randomly built decision trees is used to obtain some average classification or regression measures.

Suppose that instance $\mathbf{x} \in \mathbb{R}^m$ has to be interpreted under condition that the black-box model has been trained on the dataset $D = \{(\mathbf{x}_1, y_1), \ldots, (\mathbf{x}_n, y_n)\}$. A general scheme of the first approach called ensemble of random SHAPs (ER-SHAP) for case $N = 3$ is illustrated in Figure 1. ER-SHAP is iteratively constructed by random selection of $t$ different features $N$ times. Value $t$ is a training parameter. If we refer to random forests, then one of the heuristics is $t \approx \sqrt{m}$. However, the optimal $t$ is obtained by considering many of its values. Suppose that indices of selected features at the $k$-th iteration form the set $J_k = (i_1, \ldots, i_t)$. The corresponding vector of $t$ features is regarded as an instance $\mathbf{z}_k = (x_{i_1}, \ldots, x_{i_t}) \in \mathbb{R}^t$. Subsets of selected features with indices $J_k$ are shown in Figure 1 as successive features. However, this is only a schematic illustration. Features are randomly selected in accordance with the uniform distribution and can be located at arbitrary places of vector $\mathbf{x}$.

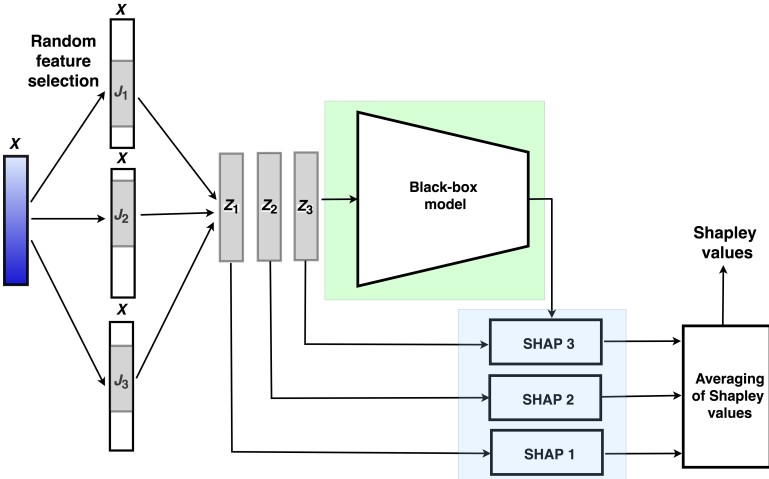

**Figure 1.** A scheme of the ER-SHAP.

As a result, we have a set of $N$ instances $\mathbf{z}_1, \ldots, \mathbf{z}_N$. The next step is to use the black-box model and SHAP to compute Shapley values for every instance such that the subset $S_k = \{\phi_i^{(k)} : i \in J_k\}$ of the Shapley values $\phi_i^{(k)}$ is produced for instance $\mathbf{z}_k$. Repeating this procedure $N$ times, we obtain a set $\mathcal{S} = \{S_1, \ldots, S_N\}$ of the Shapley values corresponding to all $\mathbf{z}_k$, $k = 1, \ldots, N$, or all input subsets of features. Having set $\mathcal{S}$, we can apply several combination rules to combining subsets $S_k$ from $\mathcal{S}$. One of the simplest rules is based on averaging of the Shapley values over all subsets $S_k$:

$$\phi_i = \frac{1}{N_i} \sum_{k:i \in J_k} \phi_i^{(k)}, \ i = 1, \ldots, m, \tag{4}$$

where $N_i$ is the number of the $i$-th feature selections among all iterations, i.e., $N_i = \sum_{k:i \in J_k} 1$.

It should be noted that the input of the black-box model has to have $m$ features. Therefore, for performing SHAPs with every $\mathbf{z}_k$, average values of features over all dataset $D$ are used to fill $m - t$ remaining features, though other methods [13] can also be used to fill these features.

Algorithm 1 can be viewed as a formal scheme implementing ER-SHAP. It is supposed that the black-box model has been already trained.

---

**Algorithm 1** ER-SHAP

---

**Require:** Training set $D$; point of interest $\mathbf{x}$; the number of iterations $N$; the number of selected features $t$; the black-box model for explaining $f(\mathbf{x})$

**Ensure:** The Shapley values $S = \{\phi_1, \dots, \phi_m\}$

  1: **for** $k = 1, k \le N$ **do**

  2:      Select randomly $t$ features from $\mathbf{x}$ and form the set $J_k$ of indices of randomly selected features $x_i$, $i \in J_k$

  3:      Use SHAP for computing $\phi_i^{(k)}$, $i \in J_k$ and form the set $S_k = \{\phi_i^{(k)} : i \in J_k\}$

  4: **end for**

  5: Combine sets $S_k$, $k = 1, \dots, N$, to compute $S$, for example, by using a simple averaging:

      $\phi_i = N_i^{-1} \sum_{k:i \in J_k} \phi_i^{(k)}$, where $N_i = \sum_{k:i \in J_k} 1$.

---

If the number of subsets $S$ in the standard SHAP or the number of differences $f(S \cup \{i\}) - f(S)$ which have to be computed is $2^m$, then the number of the same differences in ER-SHAP is $N \cdot 2^t$. For comparison purposes, if we consider a dataset with $m = 25$ and $t = \sqrt{m} = 5$, then $N$ can be taken $2^{25}/2^5 = 2^{20}$ in order to make equal computational complexity of SHAP and ER-SHAP.

*4.2. Ensemble of Random Weighted SHAPs*

The next algorithm is called the ensemble of random weighted SHAPs (ERW-SHAP) algorithm and differs from ER-SHAP in the following parts. A general scheme is shown in Figure 2. First of all, $N$ points $\mathbf{h}_1, \dots, \mathbf{h}_N$ are generated in the neighborhood of explained instance $\mathbf{x}$. These points do not need to belong to the dataset $D$. Then, $t$ features are randomly selected from every $\mathbf{h}_k$, and they produce instances $\mathbf{z}_1, \dots, \mathbf{z}_N$. Moreover, the weight $w_k$ of each instance $\mathbf{h}_k$ is defined as a function of the distance $d_k$ between the explained instance $\mathbf{x}$ and the generated neighbor $\mathbf{h}_k$. The weights are used to implement the weighted average of the Shapley values. The final Shapley values are calculated now as follows:

$$\phi_i = \frac{1}{W_i} \sum_{k:i \in J_k} w_k \phi_i^{(k)}, \ i = 1, \dots, m, \tag{5}$$

where $W_i = \sum_{k:i \in J_k} w_k$.

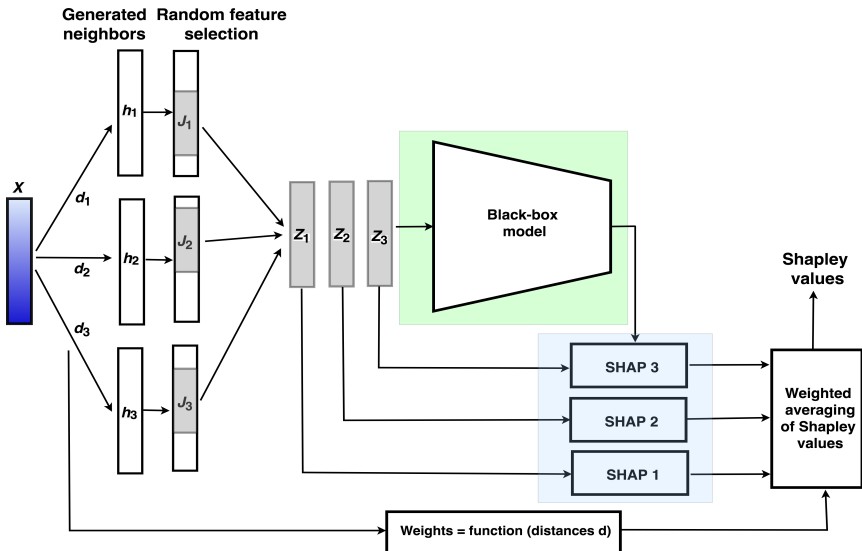

**Figure 2.** A scheme of the ERW-SHAP.

On the one hand, using these changes of ER-SHAP, we implement an idea of some kind of diversity of SHAPs to make the randomly selected feature vectors more independent. On the other hand, the approach is similar to the LIME method where the analyzed instance

is perturbed in order to build an approximating linear model around the instance to be explained. The diversity of SHAPs is a very important peculiarity of the proposed ER-SHAP. It prevents SHAP from the situation when a rule for filling the removed features produces features coinciding with the explained instance features. In this case, the Shapley values are incorrectly computed. The use of generated neighbors allows us to avoid this case and to obtain more accurate results.

Algorithm implementing ERW-SHAP differs from the similar Algorithm 1 implementing ER-SHAP only in two lines. First, after line 1 or before line 2, the line indicating how to generate neighbors has to be inserted. Second, line 5 (combination of the Shapley values) is replaced with expression (5).

*4.3. Ensemble of Random SHAPs Generated by the Random Forest*

In order to control the process of the random feature selection, it is reasonable to choose features for producing $\mathbf{z}_1, \ldots, \mathbf{z}_N$ in accordance with some probability distribution different from the uniform distribution, which would take into account the preliminary importance of features. The intuition behind this modification is to reduce the selection of unimportant features which do not impact on the black-box prediction corresponding to $\mathbf{x}$ a priori.

One of the ways to implement this control is to compute the preliminary feature importance by means of the random forest. Although it is known that the random forest does not always give acceptable results related to the feature selection problem, the proposed approach does not have this drawback because we propose to train the random forest on instances generated in the neighborhood of explained instance $\mathbf{x}$. The next algorithm is called the ensemble of random SHAPs generated by the random forest (ER-SHAP-RF) algorithm. The random forest plays a role of the important feature selection model. It can be also viewed as some kind of pre-training for important features. The idea to train the random forest on generated neighbors allows us to implement a preliminary explanation method. It should be noted that the random forest is not a unique model for selecting important features. There are many methods [59], which could be used for solving this task. We use the random forest as one of the popular and simple methods having a few parameters. In the same way, the linear regression model could be used instead of the random forest. The random forest can be used as an explanation model by applying an approach proposed by [60] based on a scalable method for transforming a decision forest into a single decision tree which is interpretable.

The LIME method can be also applied to obtain the probability distribution of features. In the case of its use, normalized absolute values of linear regression coefficients can be regarded as the probability distribution of features.

For solving the feature selection task by random forests, we use the well-known simple method [19]. According to this method, for every tree from the random forest, we compute how much the impurity is decreased by a feature. The more the feature decreases the impurity, the more important the feature is. The impurity decreasing is averaged across all trees in the random forest, and the obtained value corresponds to the final importance of the feature.

The proposed approach may lead to small probabilities of unimportant features. However, it does not mean that these features will not selected for using in an explanation by means of SHAP. They have a smaller chance to be selected under condition that their probabilities are not equal to zero. This implies that the classification or regression models for constructing the probability distribution $P$ should not provide some sparse predictions such as the Lasso because only a small part of features in this case will take part in explanation.

A general scheme of ER-SHAP-RF is shown in Figure 3 where a number, say $M$, of neighbors $\mathbf{h}_1, \ldots, \mathbf{h}_M$ are generated around the instance $\mathbf{x}$ to be explained. Every generated neighbor $\mathbf{h}_j$ is fed into the black-box model to obtain its class label $y_j^*$. It should be noted

that the training instances can be taken as neighbors. However, they should be classified by using the black-box model in order to take into account this model in an explanation.

Having points $(\mathbf{h}_j, y_j^*)$, we train the random forest which provides a feature importance measure in the form of the probability distribution $P = (p_1, \ldots, p_m)$. The distribution $P$ is used to select features from instance $\mathbf{x}$ for constructing the vectors $\mathbf{z}_1, \ldots, \mathbf{z}_N$, namely $t$ features are selected from $\mathbf{x}$ with replacement $N$ times in accordance with the distribution $P$. SHAPs are used to find the Shapley values of vectors $\mathbf{z}_1, \ldots, \mathbf{z}_N$. They are combined similarly to ERW-SHAP by means of averaging as follows:

$$\phi_i = \frac{1}{N_i} \sum_{k:i \in J_k} \phi_i^{(k)}, \ i = 1, \ldots, m. \tag{6}$$

It is important that the number $N_i$ of the $i$-th feature selections among all iterations $N$ is used instead of $N$.

The whole algorithm can be divided into two stages which are separated in time. According to the first stage, neighbors $\mathbf{h}_1, \ldots, \mathbf{h}_M$ are generated for obtaining predictions $y_1^*, \ldots, y_M^*$ by the black-box model and for training the random forest which provides probabilities of features. This stage is depicted by dashed lines in Figure 3. The second stage is to use these probabilities for using SHAPs. This stage is depicted by solid lines in Figure 3.

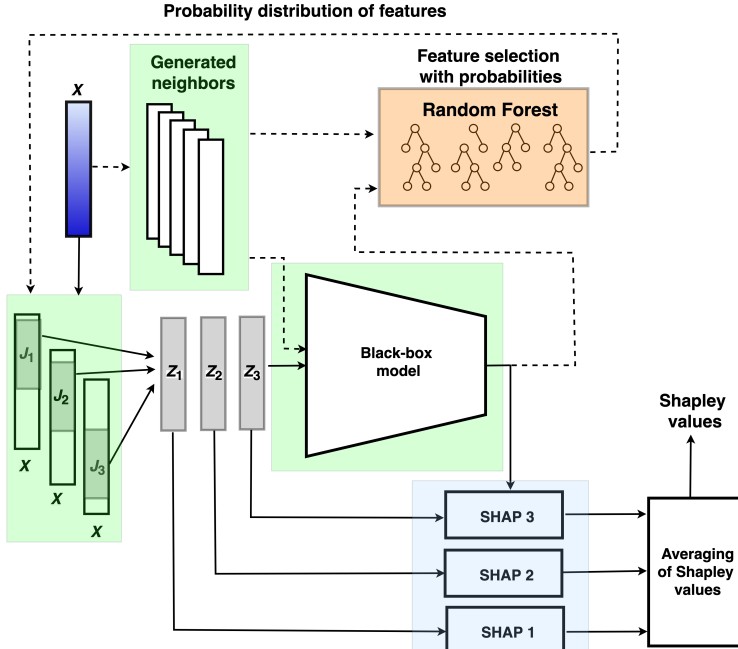

**Figure 3.** A scheme of the ER-SHAP-RF.

The random forest should be built with a large depth of trees and with a small number of trees in order to avoid a rather sparse probability distribution of features when a large part of probabilities will be equal to zero or close to zero. Another way for avoiding small probabilities of features is to apply calibration methods and to recalculate the obtained probabilities, for example, by using the temperature scaling as the simplest extension of Platt scaling [61]:

$$p_k^* = \frac{\exp(p_k/T)}{\sum_{i=1}^m \exp(p_i/T)}, \ k = 1, \ldots, m, \tag{7}$$

where $T$ is the temperature which controls the smoothness of the probability distribution, but it does not change the relationship of probabilities $p_k$, $i = 1, \ldots, m$.

Algorithm 2 implementing ER-SHAP-RF can be viewed as an extension of ER-SHAP.

---

**Algorithm 2** ER-SHAP-RF

---

**Require:** Training set $D$; point of interest $\mathbf{x}$; the number of iterations $N$; the number of selected features $t$; the black-box model for explaining $f(\mathbf{x})$; parameters of the random forest (the number and depth of trees, number of instances for building trees)

**Ensure:** The Shapley values $S = \{\phi_1, \dots, \phi_m\}$

1: Generate $M$ instances $\mathbf{h}_1, \dots, \mathbf{h}_M$ which are from the neighborhood of $\mathbf{x}$ or from the whole training set

2: Compute the class label $y_j^* = f(\mathbf{h}_j)$ for every generated instance by using the black-box model

3: Train the random forest on $(\mathbf{h}_j, y_j^*)$, $j = 1, \dots, M$

4: Compute the probability distribution $P$ of features by using the random forest

5: **for** $k = 1, k \leq N$ **do**

6: Select randomly $t$ features from $\mathbf{x}$ in accordance with the probability distribution $P$ and form the index set $J_k$ of features

7: Use SHAP for computing $\phi_i^{(k)}$, $i \in J_k$ and form the set $S_k = \{\phi_i^{(k)} : i \in J_k\}$

8: **end for**

9: Combine sets $S_k$, $k = 1, \dots, N$, to compute $S$, for example, by using a simple averaging: $\phi_i = N_i^{-1} \sum_{k: i \in J_k} \phi_i^{(k)}$, where $N_i = \sum_{k: i \in J_k} 1$.

---

It is interesting to point out that the fourth algorithm can also be proposed, which is represented as a combination of ERW-SHAP and ER-SHAP-RF. $N$ points are generated for implementing diversity in accordance with ERW-SHAP, and $M$ points are generated for training the random forest in accordance with ER-SHAP-RF and for computing the prior probability distribution $P$ of features. Then, the random features are selected not from the vector $\mathbf{x}$, as it is done in ER-SHAP-RF, but from every vector $\mathbf{h}_k$ with the probability distribution $P$, $k = 1, \dots, N$. However, this algorithm is not studied because it can be regarded as the combination of ERW-SHAP and ER-SHAP-RF, which are analyzed in detail.

Let us consider complexity of the models. If we assume that the complexity of the black-box model is $B(m, n)$, the random forest tree depth is $d$, and the number of trees is $T$, then the complexity of the random forest training is $O(T \cdot m \cdot M \cdot \log(M))$, the complexity of the random forest predicting is $O(T \cdot d \cdot M)$. The complexity of SHAP is $O(2^m \cdot B(m, n))$. The complexity of ER-SHAP is $O(2^t \cdot N \cdot B(m, n))$. It follows from the above that ER-SHAP is more effective than SHAP when $2^m > 2^t \cdot N$ or $m > t + \log_2(N)$. The complexity of ER-SHAP-RF is

$$O(2^t \cdot N \cdot B(m, n) + T \cdot m \cdot M \cdot \log(M) \cdot B(m, n) + T \cdot d \cdot M).$$

It can be seen from the above that the complexity of the random forest training and predicting does not sufficiently impact on the complexity of ER-SHAP-RF in comparison with the complexity of ER-SHAP. The same can be said about ERW-SHAP.

## 5. Numerical Experiments

First, we consider several numerical examples for which training instances are randomly generated. Each generated synthetic instance consists of 5 features. Two features are generated as shown in Figure 4, and other features are uniformly generated in intervals $[-1, 1]$. Each picture in Figure 4 corresponds to a certain location of instances of two classes such that the instances of classes 0 and 1 are depicted by small triangles and crosses, respectively. This generation corresponds to the case when the first two features may be important. These features allow us to analyze the feature importance in accordance with the data location and with the separating function. Other features are not important, and they are used to generalize numerical experiments with synthetic data.

Separating functions in Figure 4 are obtained by means of SVM which can be regarded as the black-box model. It used the RBF kernel whose parameter depends on a dataset trained. The SVM allows us to obtain different separating functions by changing the kernel

parameter. Figure 4a illustrates the linearly separating case. The specific class area in the form of a stripe is shown in Figure 4b. A saw-based separating function is used in Figure 4c. The class area in the form of a wedge is given in Figure 4d. A checkerboard with an attempt of SVM to separate the checkerboard cages can be found in Figure 4e. For every generated dataset from Figure 4, we compare SHAP with the proposed modifications.

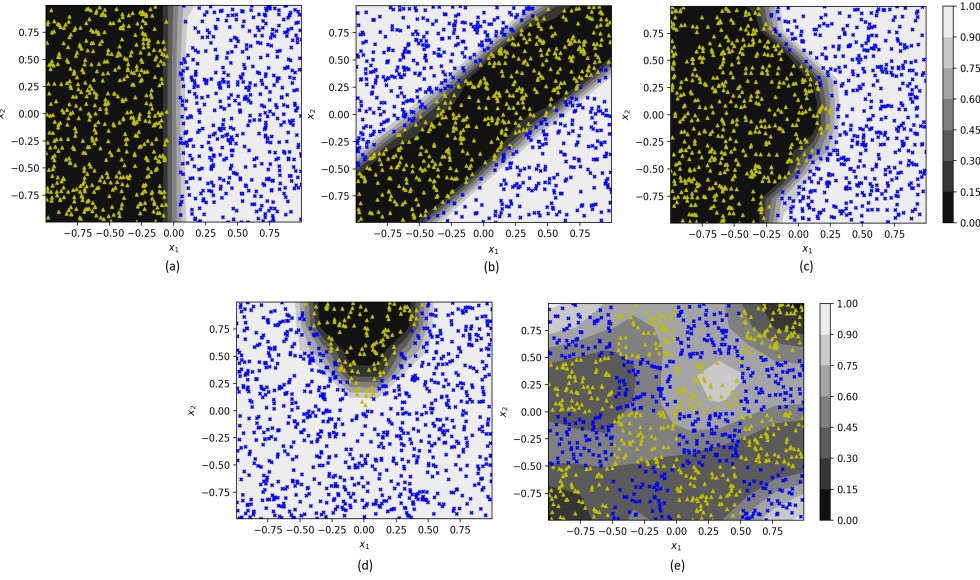

**Figure 4.** Five synthetic datasets and the boundaries between classes provided by SVM in the form of: (**a**) the linear separation; (**b**) a stripe; (**c**) a saw-based separating function; (**d**) a wedge; (**e**) the checkerboard cages.

**Measures for comparison**: In order to compare the proposed modifications with the original SHAP method, we use the concordance index $C$ of pairs, which is defined as the proportion of concordant pairs of the Shapley values divided by the total number of possible evaluation pairs. Let $\phi_i^*$ and $\phi_i$ be the Shapley values obtained by means of the original SHAP method and one of its modifications (ER-SHAP, ERW-SHAP, ER-SHAP-RF), respectively. Two pairs of the Shapley values $(\phi_i, \phi_j)$ and $(\phi_i^*, \phi_j^*)$ are concordant if they hold $(\phi_i > \phi_j, \phi_i^* > \phi_j^*)$ or $(\phi_i < \phi_j, \phi_i^* < \phi_j^*)$. In contrast to the well-known C-index in survival analysis, the introduced concordance index compares predictions provided by two methods. If the index is close to 1, then the models provide the same results. A motivation for the concordance index introduction is that the Shapley values computed by using original SHAP and the proposed modifications may be different. However, we are interested in their relationship. If the original SHAP method gives the inequality $\phi_i^* > \phi_j^*$ for some $i$ and $j$, then we are expecting to have $\phi_i > \phi_j$ for the proposed method, but not equalities $\phi_i^* = \phi_i$ and $\phi_j^* = \phi_j$. It should be noted that original SHAP may provide incorrect results. Therefore, the introduced concordance index should be viewed as a desirable measure under condition of correct SHAP results.

We use the Kernel SHAP [10] in numerical experiments and compare obtained results with it.

In spite of importance of the concordance index, we also use the normalized Euclidean distance $E$ between vectors $(\phi_1^*, \ldots, \phi_m^*)$ and $(\phi_1, \ldots, \phi_m)$. The distance shows how the absolute Shapley values of two methods are close to each other. It is important to take into account that the Shapley values in the original SHAP method satisfy the efficiency property when $\phi_1^* + \ldots + \phi_m^* = f(\mathbf{x}) - f(\varnothing)$. This property is not fulfilled for modifications because they do not enumerate all subsets of features. Therefore, in order to consider the Shapley values in the same scale, all values $\phi_i$ and $\phi_i^*$ are normalized to be in interval $[0, 1]$.

### 5.1. ER-SHAP

First, we consider the results of the numerical experiments obtained by means of the ER-SHAP with the SVM as a black-box model trained on the datasets shown in Figure 4. The explained instance for the experiments has all identical features which are equal to 0.25. The concordance indices of the ER-SHAP as functions of the number of iterations $N$ for the numbers of the selected features $t = 2$ (the solid line) and $t = 3$ (the dashed line) are illustrated in Figure 5, where pictures (a–e) correspond to pictures (a–e) shown in Figure 4. It can be seen from Figure 5 that the concordance index increases with $N$ on average. This implies that the ER-SHAP provides results comparable with the SHAP. It can be also seen from the pictures that the concordance index is significantly larger for $t = 3$ in comparison to the case of $t = 2$. This observation is obvious because the large number of selected features in each iteration brings the modification closer to the original SHAP method. Though, one can see from Figure 5b that the case $t = 2$ provides better concordance index by $N \geq 7$.

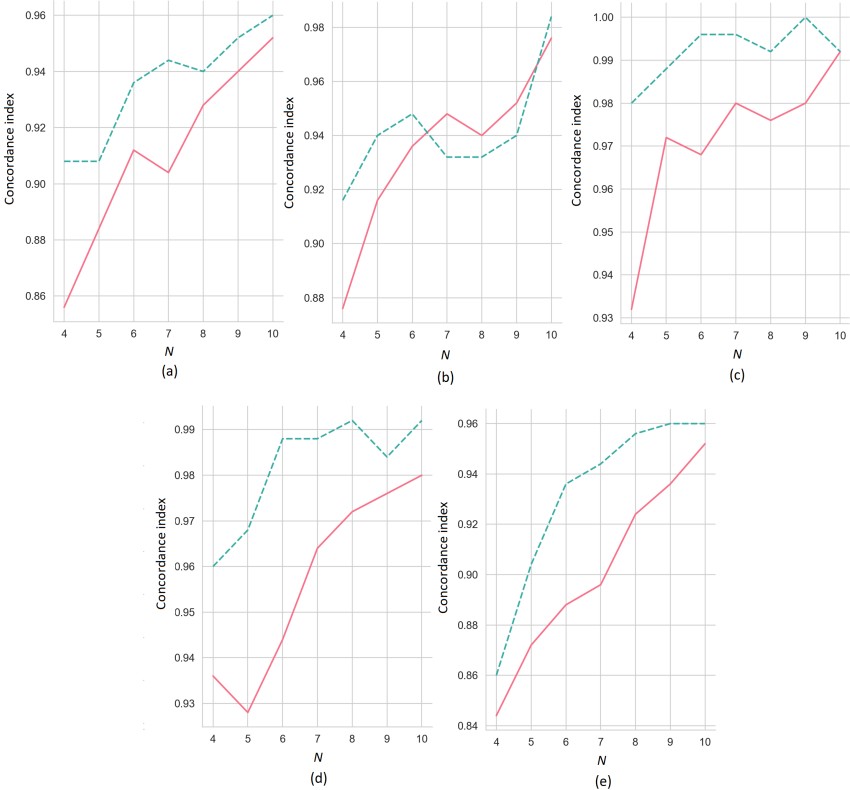

**Figure 5.** Concordance indices of ER-SHAP as functions of the number of iterations $N$ for $t = 2$ (the solid line) and 3 (the dashed line) for trained SVMs and five datasets depicted in corresponding Figure 4a–e.

Figure 6 illustrates how the Euclidean distances between the ER-SHAP and SHAP as functions of the number of iterations $N$ for $t = 2$ (the solid line) and 3 (the dashed line) decrease with $N$. We again consider five training sets, shown in Figure 4.

In order to explicitly illustrate how the Shapley values $\phi_i^*$ and $\phi_i$ obtained by the SHAP and ER-SHAP, respectively, are close to each other, we show the Shapley values for all five cases in Figure 7. It can be seen that despite the difference in the absolute values, the Shapley values indicate to the same important features.

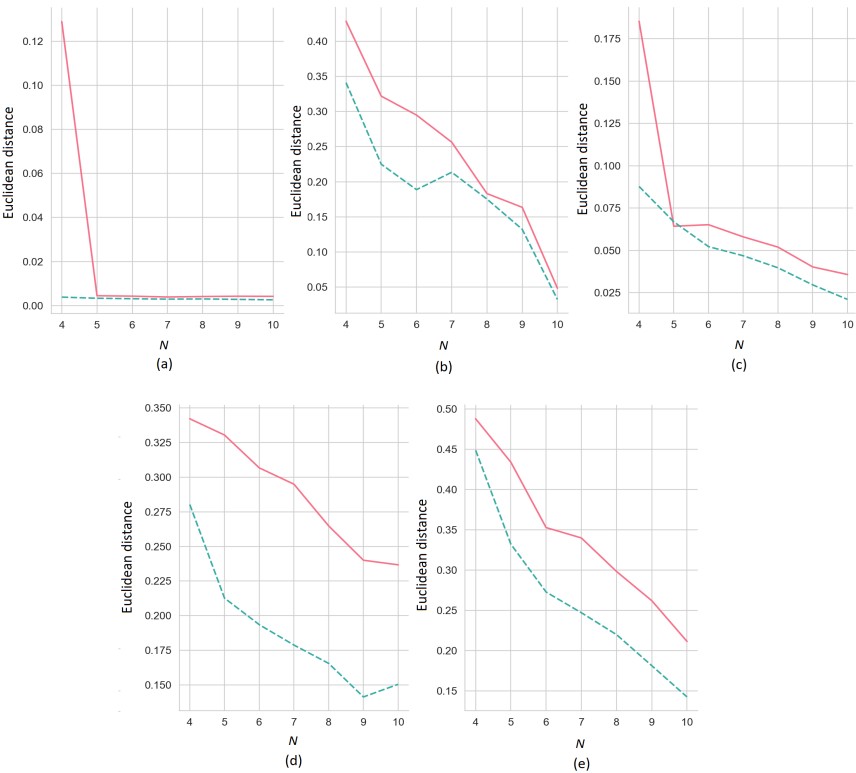

**Figure 6.** Euclidean distances between ER-SHAP and SHAP as functions of the number of iterations $N$ for $t = 2$ (the solid line) and 3 (the dashed line) for trained SVMs and five datasets depicted in corresponding Figure 4a–e.

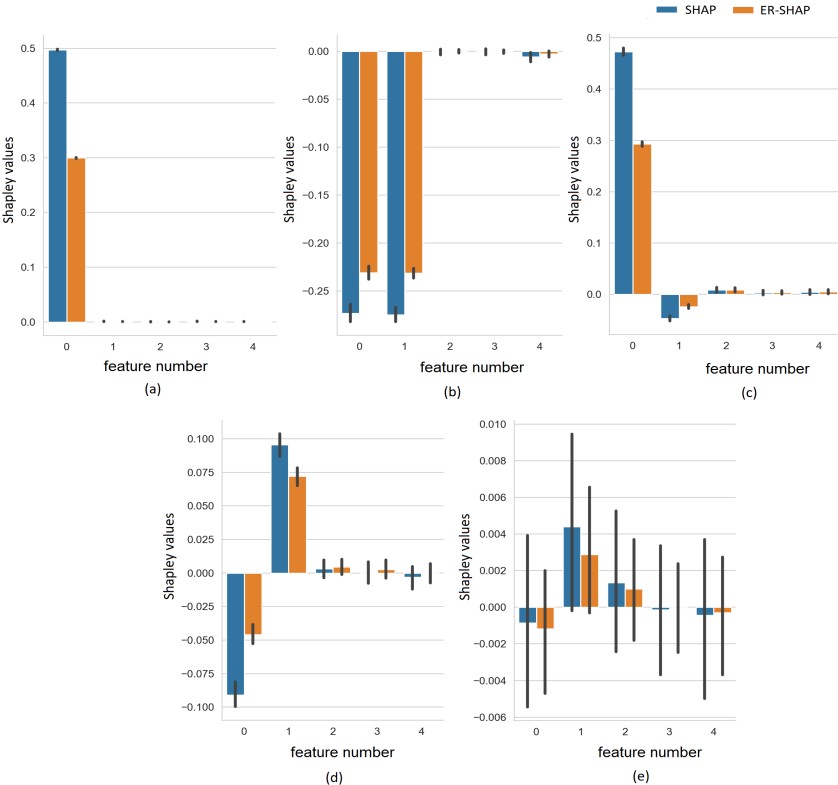

**Figure 7.** Shapley values obtained by means of SHAP and ER-SHAP for all features of five datasets depicted in corresponding Figure 4a–e and trained SVMs as black-boxes.

### 5.2. ERW-SHAP

To study the ERW-SHAP, the features of the explained instance are noised by using the normal distribution of noise with the zero expectation and standard deviations 0.01 and 0.1. The weights of the generated instances $\mathbf{h}_i$ are defined by

$$w_i = \exp\left(-\|\mathbf{h}_i - \mathbf{x}\|^2\right). \tag{8}$$

We consider the similar results of the numerical experiments obtained by means of the ERW-SHAP with the SVM as a black-box model trained on the datasets shown in Figure 4 with the same explained instance. The concordance indices of the ERW-SHAP as functions of $N$ for $t = 2$ (the solid line) and $t = 3$ (the dashed line) are illustrated in Figure 8, where pictures (a–e) correspond to pictures (a–e) shown in Figure 4. The standard deviation of the normal distribution generating noise is 0.01. If we compare the concordance indices for the ERW-SHAP (Figure 8) and for the ER-SHAP (Figure 5), then it is obvious that the ERW-SHAP provides better results in comparison to the ERW-SHAP for most of the datasets.

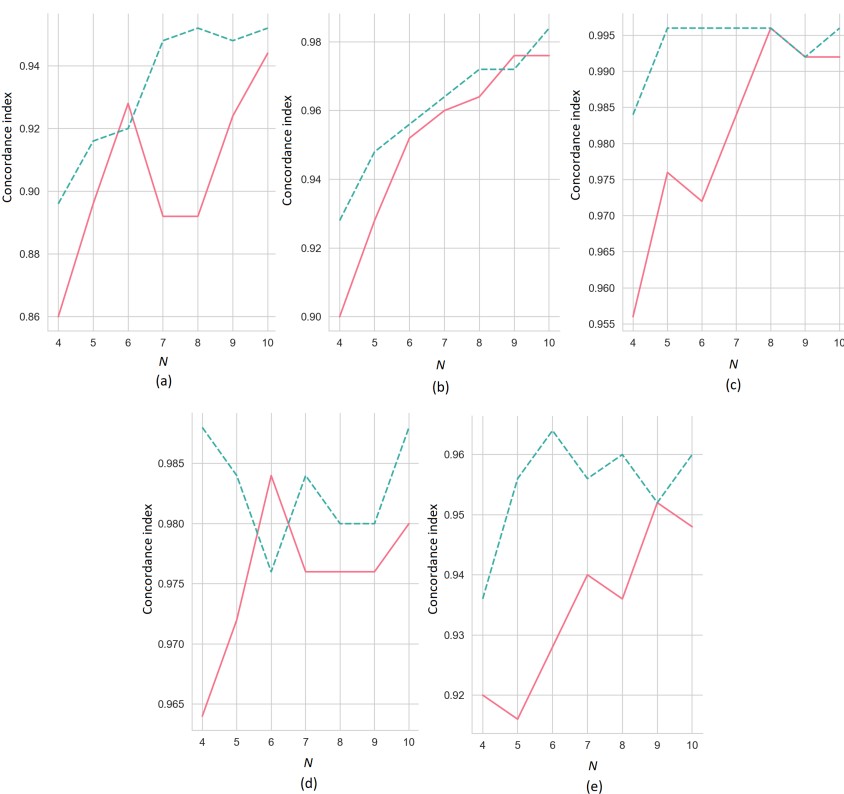

**Figure 8.** Concordance indices of ERW-SHAP as functions of $N$ for $t = 2$ (the solid line) and 3 (the dashed line) and for five datasets depicted in corresponding Figure 4a–e and trained SVMs.

At the same time, the Euclidean distances between the SHAP and ERW-SHAP slightly differ from the same distances between the SHAP and ER-SHAP. This follows from Figure 9 where the Euclidean distances between the ERW-SHAP and SHAP as functions of $N$ for $t = 2$ (the solid line) and 3 (the dashed line) are presented for the above datasets.

To illustrate how the Shapley values $\phi_i^*$ and $\phi_i$ obtained by the SHAP and ERW-SHAP, respectively, are close to each other, we show the Shapley values for the five cases in Figures 10 and 11. Figures 10 and 11 provide results under the condition that the normal distribution of the generated noise has the standard deviations 0.1 and 0.01, respectively. We again observe that the ERW-SHAP can be regarded as a good approximation of the SHAP because the Shapley values of the ERW-SHAP and SHAP are very close to each other.

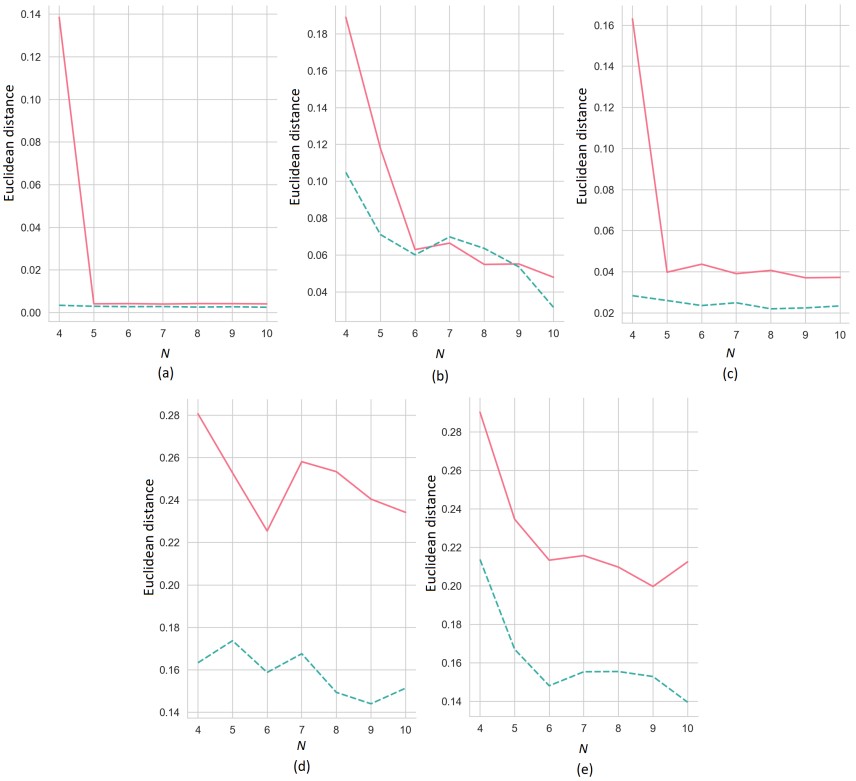

**Figure 9.** Euclidean distances between ERW-SHAP and SHAP as functions of $N$ for $t = 2$ (the solid line) and 3 (the dashed line) for five datasets depicted in corresponding Figure 4a–e and trained SVMs as black-boxes.

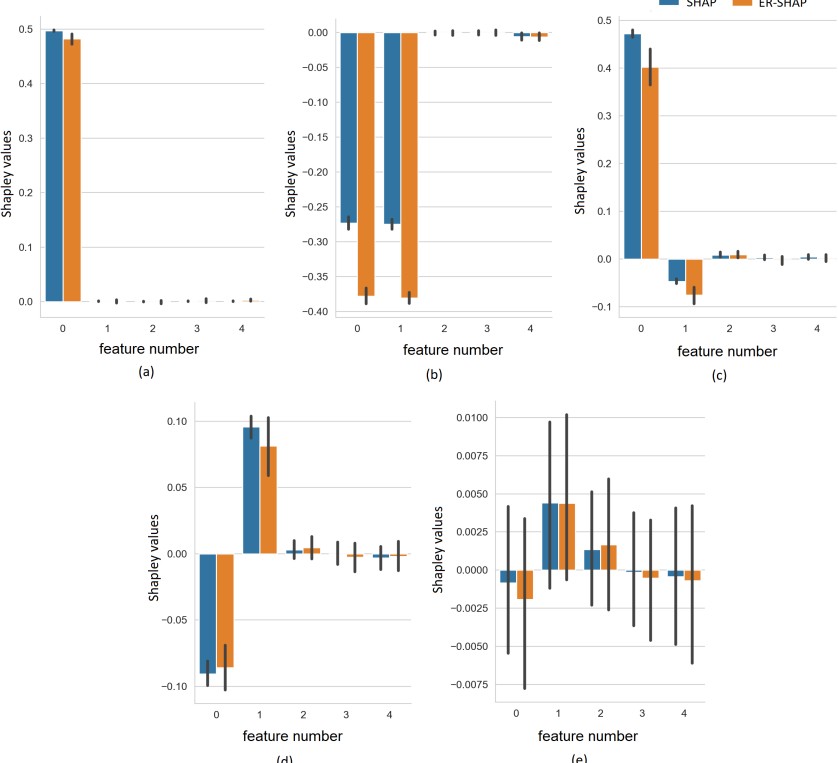

**Figure 10.** Shapley values obtained by means of SHAP and ERW-SHAP for all features of five datasets depicted in corresponding Figure 4a–e and trained SVMs under condition of using the normal distribution of feature changes with the standard deviation 0.1

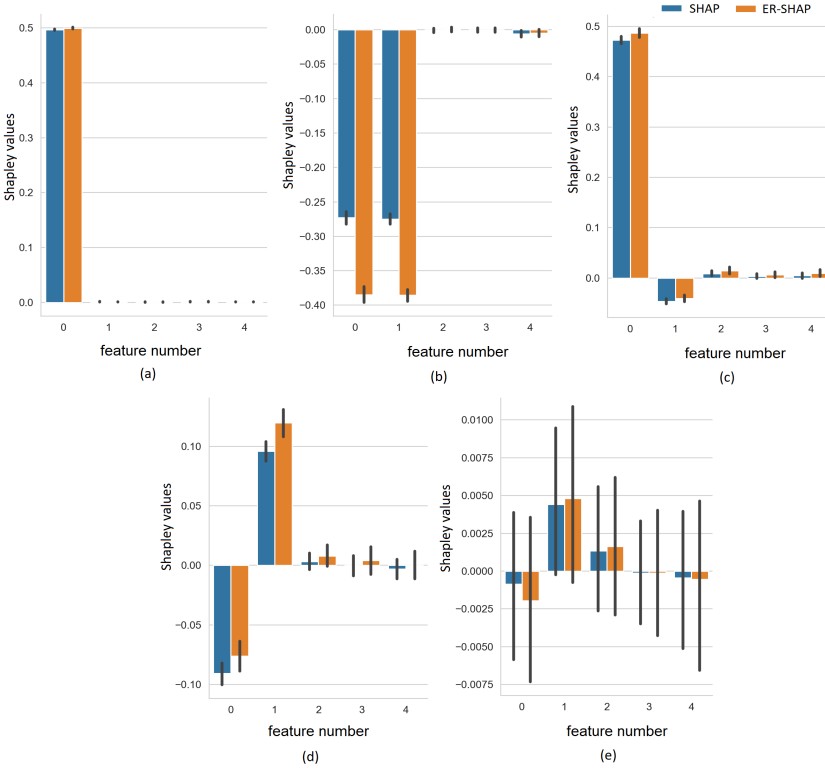

**Figure 11.** Shapley values obtained by means of SHAP and ERW-SHAP for all features of five datasets depicted in corresponding Figure 4a–e and trained SVMs under condition of using the normal distribution of feature changes with the standard deviation 0.01.

*5.3. ER-SHAP-RF*

We again study the modification by using the datasets shown in Figure 4. The result shows that the ER-SHAP-RF outperforms the ER-SHAP as well as the ERW-SHAP for most of the datasets. Indeed, if we compare the concordance indices for the ER-SHAP-RF (Figure 12) with the ERW-SHAP (Figure 8) and ER-SHAP (Figure 5), then we see that all the examples provide better results. In contrast to the concordance indices, the Euclidean distances shown in Figure 13 demonstrate worse results. At the same time, the Shapley values given in Figure 14 almost coincide with the corresponding values obtained by means of the ERW-SHAP (Figure 11). It should be noted that a more accurate tuning of the random forest might provide outperforming results.

Let us summarize the numerical results obtained on the synthetic data for the models ER-SHAP, ERW-SHAP, and ER-SHAP-RF. First, we compare the C-indices corresponding to the models, which are depicted in Figures 5, 8, and 12. It can clearly be seen from the results that the ER-SHAP-RF outperforms the ER-SHAP as well as the ERW-SHAP for all five datasets. The ERW-SHAP outperforms the ER-SHAP for the datasets depicted in Figure 4b,c,e. However, the ERW-SHAP is inferior to the ER-SHAP for the datasets depicted in Figure 4a,d. Moreover, Figure 8 shows that the C-index of the ERW-SHAP behaves unstably. However, if we compare the ER-SHAP and ERW-SHAP using the Euclidean distances between the results provided by these models and the SHAP, then we can conclude that the ERW-SHAP outperforms the ER-SHAP for all the datasets. It is interesting to point out that the ER-SHAP-RF outperforms the ERW-SHAP only for the first dataset (see Figure 4a) if we consider the Euclidean distances. However, we have mentioned that the Euclidean distance cannot be viewed as the best measure for a comparison of the explainable models. Therefore, we can conclude that the ER-SHAP-RF provides the best results, though this model requires generating neighbors and training the random forest.

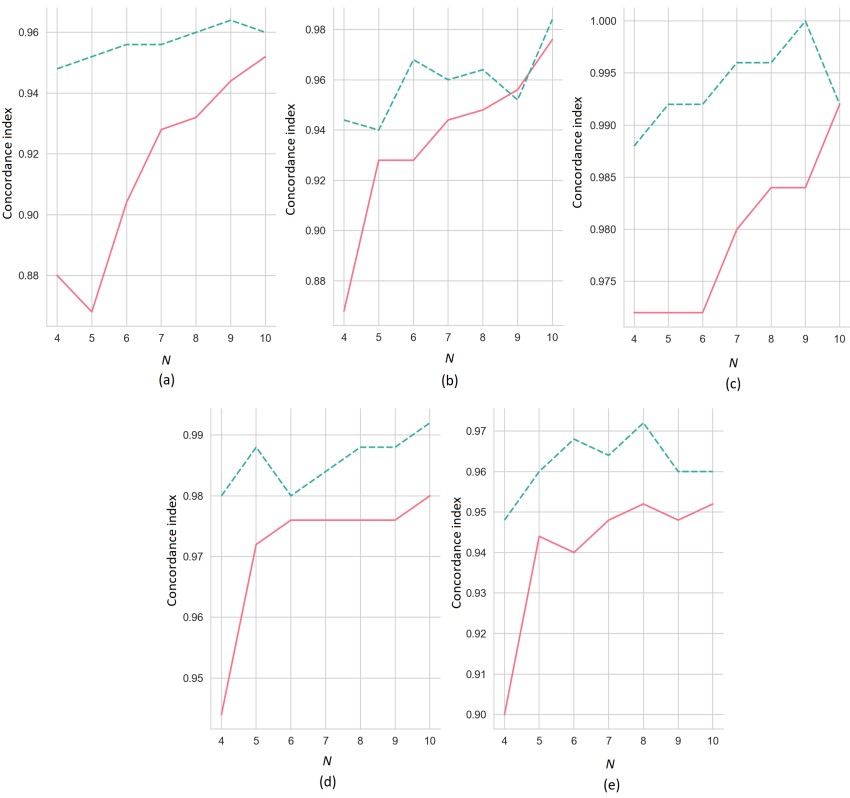

**Figure 12.** Concordance indices of ER-SHAP-RF as functions of $N$ for $t = 2$ (the solid line) and 3 (the dashed line) for five datasets depicted in corresponding Figure 4a–e and trained SVMs

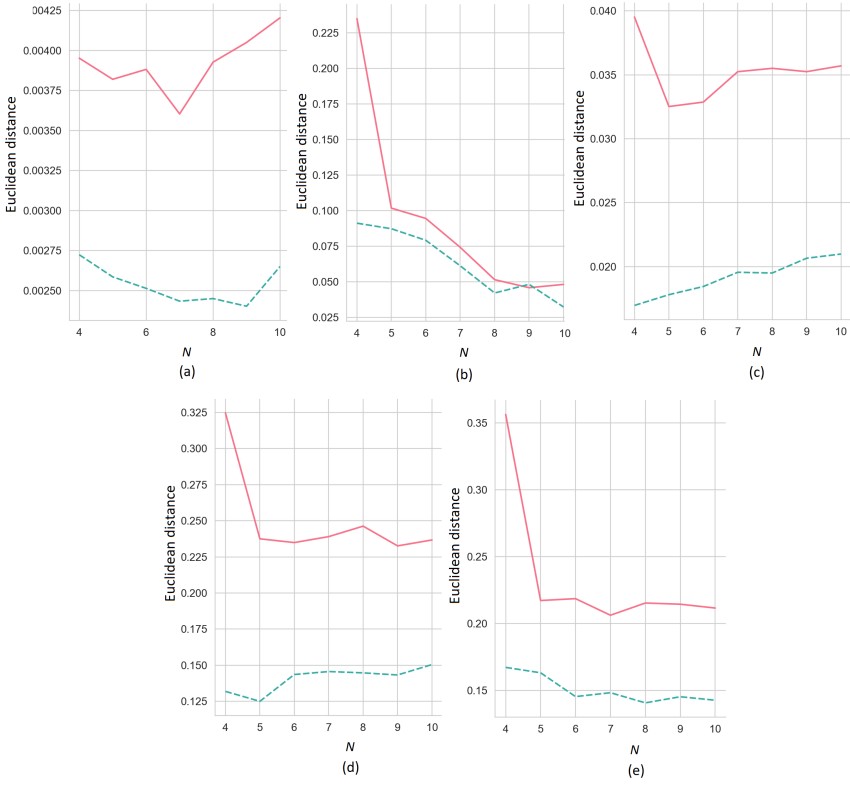

**Figure 13.** Euclidean distances between ER-SHAP-RF and SHAP as functions of $N$ for $t = 2$ (the solid line) and 3 (the dashed line) for five datasets depicted in corresponding Figure 4a–e and trained SVMs.

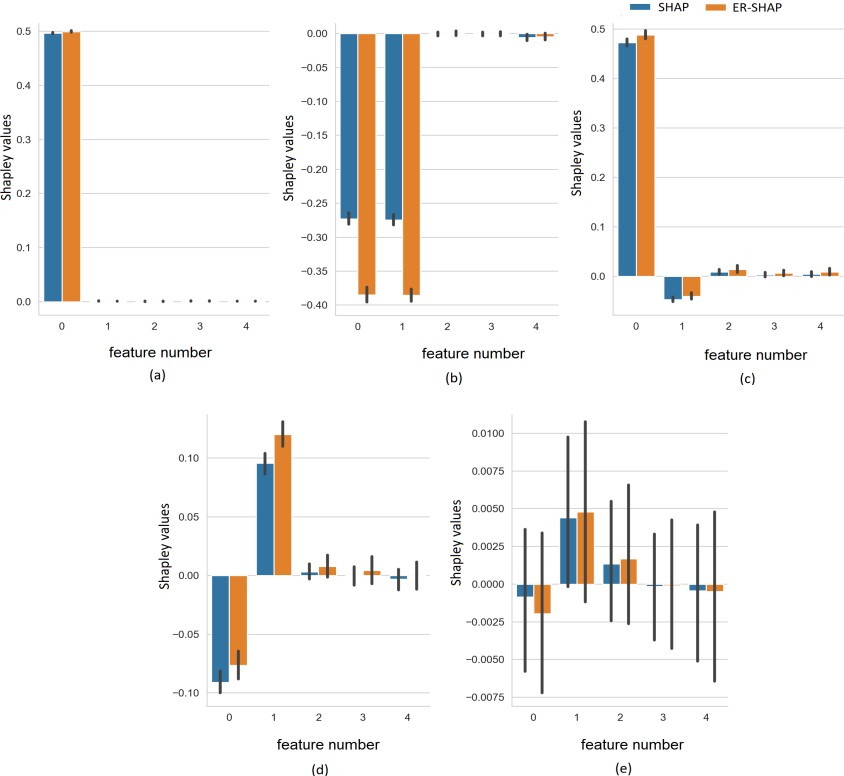

**Figure 14.** Shapley values obtained by means of SHAP and ER-SHAP-RF for all features of five datasets depicted in corresponding Figure 4a–e and trained SVMs.

*5.4. Boston Housing Dataset*

Let us consider the real data called the Boston Housing dataset. It can be obtained from the StatLib archive (http://lib.stat.cmu.edu/datasets/boston, accessed on 2 November 2022). The Boston Housing dataset consists of 506 instances such that each instance is described by 13 features.

The heatmap reflecting the concordance index of the ER-SHAP for the Boston Housing dataset is shown in Figure 15. Each element at position $(i, j)$, where $i$ and $j$ are the numbers of the row and column, respectively, indicates the value of the concordance index. Each row corresponds to the number of iterations $N$, and each column corresponds to the number of selected features $t$. It can be seen from Figure 15 that the concordance index increases with $N$ and $t$. This implies that the ER-SHAP provides results coinciding with the SHAP by rather large numbers of iterations $N$. Figure 16 illustrates how the computation time $\tau_{\text{SHAP}}$ of the SHAP exceeds the computation time $\tau_{\text{ER-SHAP}}$ of the ER-SHAP. The heatmap shows the ratio $\tau_{\text{ER-SHAP}}/\tau_{\text{SHAP}}$. One can see, from Figure 16, a clear advantage of using the ER-SHAP from the computational point of view.

Figure 17 shows the heatmap of the concordance index of the ERW-SHAP for the Boston Housing dataset. It is clearly seen from Figure 17 that the introduction of weights and generated instances significantly improves the approximation.

The Shapley values obtained by means of the ER-SHAP and SHAP as well as the ERW-SHAP and SHAP are shown in Figures 18 and 19, respectively. One can see from Figures 18 and 19 that the ERW-SHAP can be viewed as a better approximation of the SHAP because the corresponding bars almost coincide, as shown in Figure 19. It should be noted that the Shapley values provided by the ER-SHAP also behave like values of the SHAP (see Figure 18), but they do not coincide for the most important features.

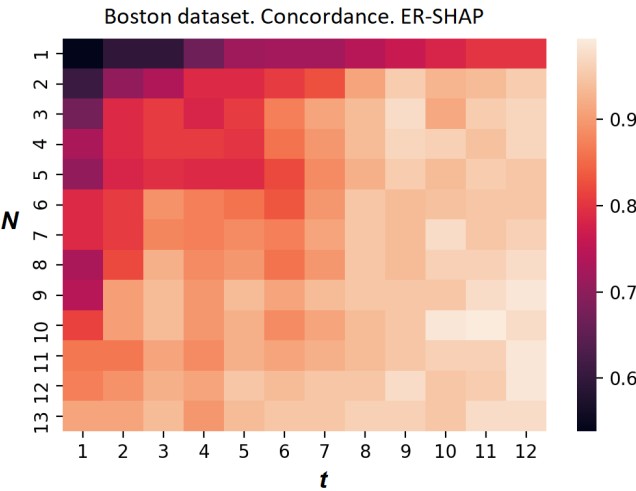

**Figure 15.** The heatmap reflecting the concordance index *C* obtained by ER-SHAP for the Boston Housing dataset.

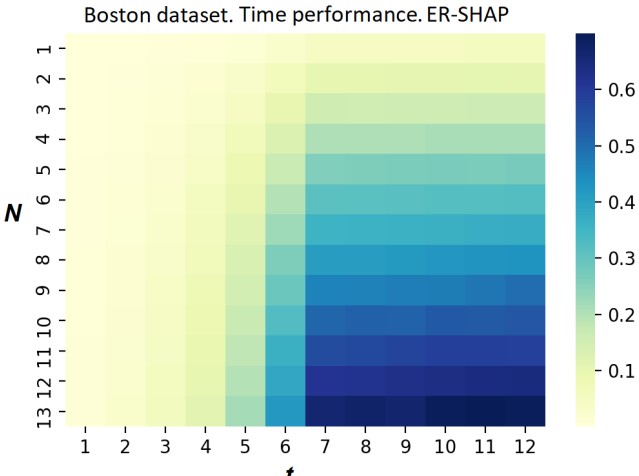

**Figure 16.** The heatmap illustrating the relationship between computation times of SHAP and ER-SHAP for the Boston Housing dataset.

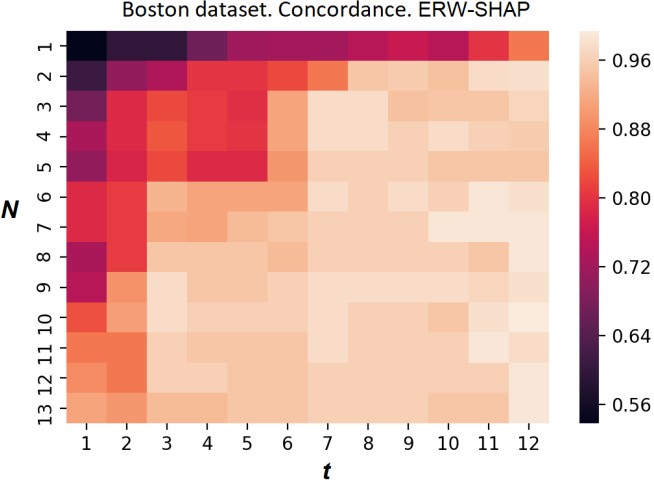

**Figure 17.** The heatmap reflecting the concordance index *C* obtained by ERW-SHAP for the Boston Housing dataset.

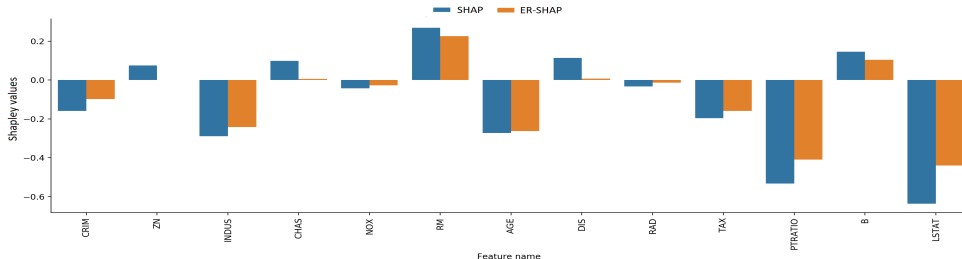

**Figure 18.** Shapley values obtained by means of SHAP and ER-SHAP for the Boston Housing dataset.

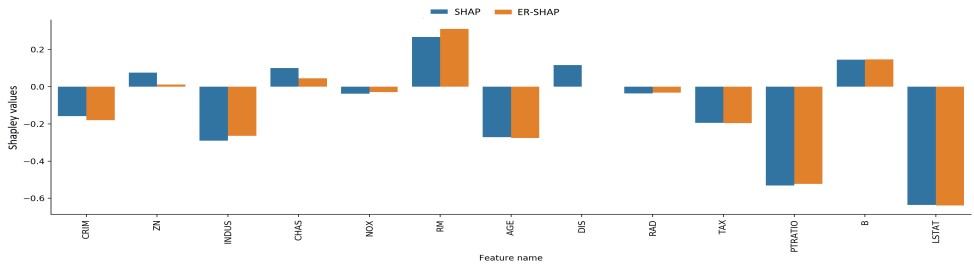

**Figure 19.** Shapley values obtained by means of SHAP and ERW-SHAP for features of the Boston Housing dataset under condition of using the normal distribution of feature changes with the standard deviation 0.1.

Figures 20 and 21 illustrate the heatmaps of the concordance index of the ER-SHAP-RF for the Boston Housing dataset. They are obtained without using the temperature scaling in accordance with (7) and with this calibration method, respectively. It is interesting to observe from Figures 20 and 21 that the use of the calibration leads to a more contrasting heatmap and to an obvious improvement in the approximation quality.

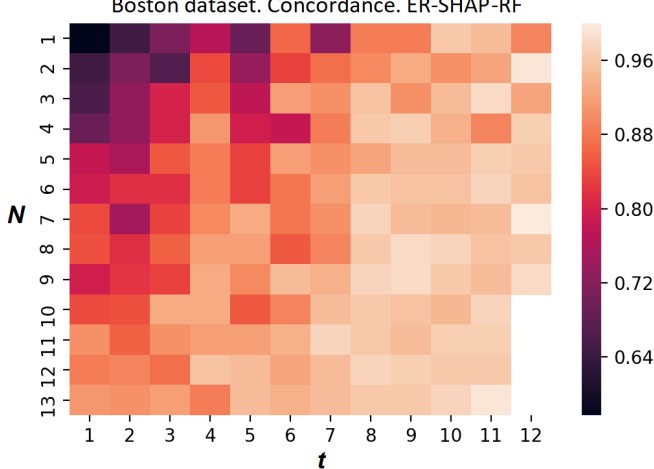

**Figure 20.** The heatmap reflecting the concordance index $C$ obtained by ER-SHAP-RF for the Boston Housing dataset without using the temperature scaling.

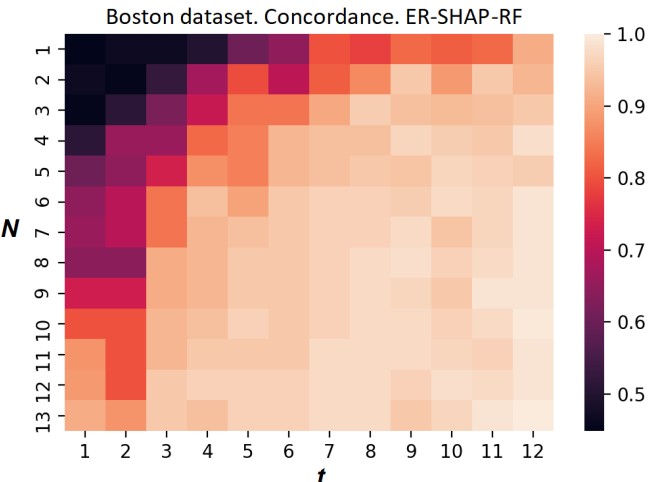

**Figure 21.** The heatmap reflecting the concordance index $C$ obtained by ER-SHAP-RF for the Boston Housing dataset using the temperature scaling.

### 5.5. Breast Cancer Dataset

The next real dataset is the Breast Cancer Wisconsin (Diagnostic) dataset. It can be found in the well-known UCI Machine Learning Repository (https://archive.ics.uci.edu, accessed on 2 November 2022). The Breast Cancer dataset contains 569 instances such that each instance is described by 30 features. For the classes of the breast cancer diagnosis, the malignant and the benign are assigned by classes 0 and 1, respectively. We consider the corresponding model in the framework of the regression with outcomes in the form of probabilities from 0 (malignant) to 1 (benign).

The heatmaps given in Figures 22 and 23 are similar to the same heatmaps obtained for the Boston Housing dataset (Figures 15 and 16). It is interesting to observe from Figure 23 that there are $N$ and $t$ such that the ratio $\tau_{\text{ER-SHAP}}/\tau_{\text{SHAP}}$ is larger 1. This implies that the SHAP is computationally simpler in comparison to the ER-SHAP. However, these cases take place only for large values $N$ and $t$.

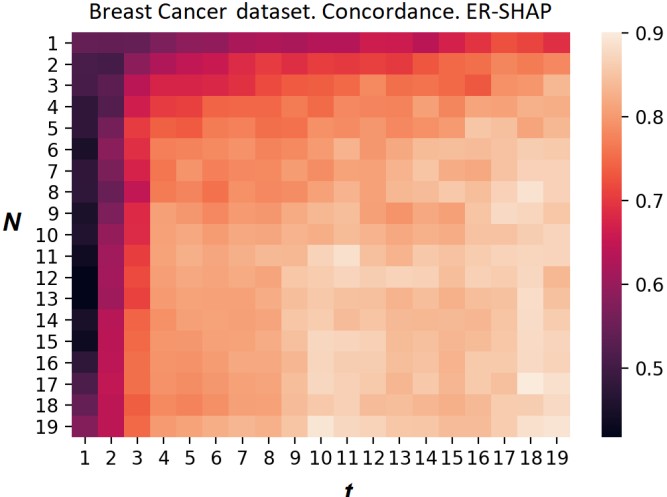

**Figure 22.** The heatmap reflecting the concordance index $C$ obtained by ER-SHAP for the Breast Cancer dataset.

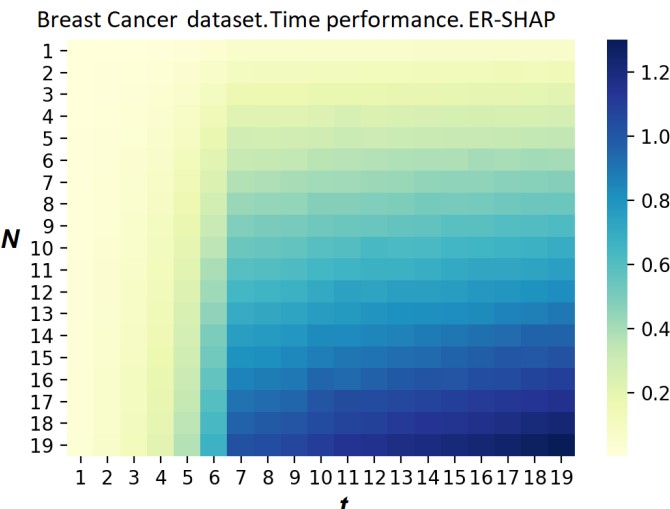

**Figure 23.** The heatmap illustrating the relationship between computation times of SHAP and ER-SHAP for the Breast Cancer Housing dataset.

At first glance, it is difficult to evaluate from Figure 24 whether the ERW-SHAP provides better results than the ER-SHAP. Figure 24 shows the heatmap of the concordance index of the ERW-SHAP for the Breast Cancer dataset. However, we can see that the legend in Figure 24 is changed in the interval $[0.4, 0.95]$, whereas the legend in Figure 22 is changed in the interval $[0.4, 0.9]$. This implies that the ERW-SHAP outperforms the ER-SHAP in this numerical example.

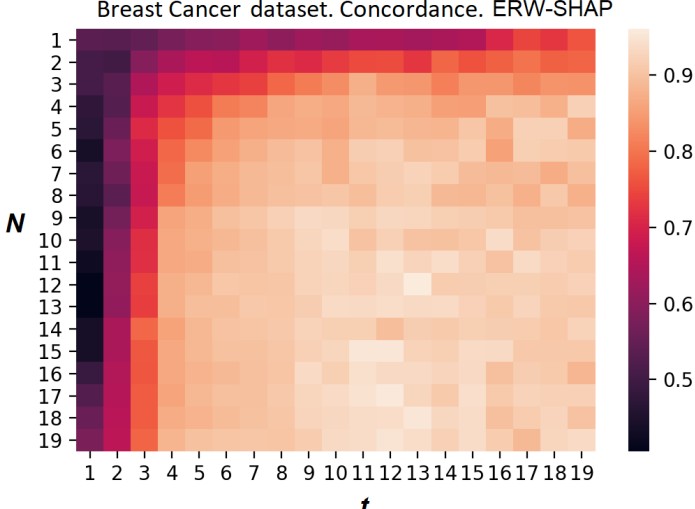

**Figure 24.** The heatmap reflecting the concordance index $C$ obtained by ERW-SHAP for the Breast Cancer dataset.

The Shapley values for all the features of the Breast Cancer dataset, which are obtained by means of the ER-SHAP and SHAP, are shown in Figure 25. The similar values obtained by means of the ERW-SHAP and SHAP are shown in Figure 26. One can see from Figures 25 and 26 that the Shapley values obtained by means of the ERW-SHAP better approximate the SHAP Shapley values. For example, if we look at the feature "worst radius", which is important due to the original SHAP method, then the ER-SHAP provides the incorrect result, whereas the ERW-SHAP is totally consistent with the SHAP.

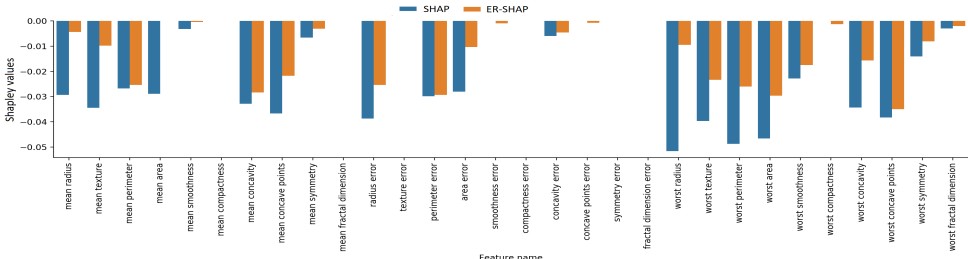

**Figure 25.** Shapley values obtained by means of SHAP and ER-SHAP for features of the Breast Cancer dataset.

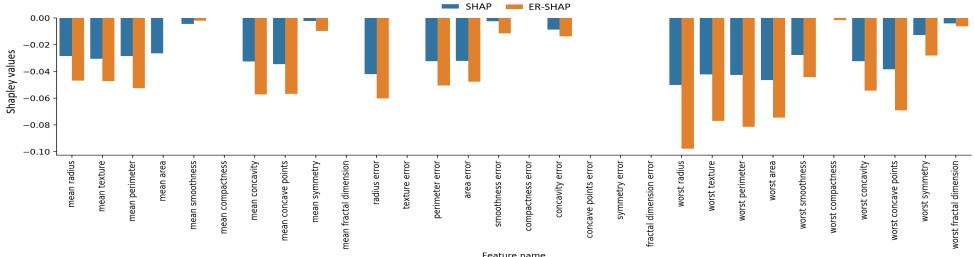

**Figure 26.** Shapley values obtained by means of SHAP and ERW-SHAP for features of the Breast Cancer dataset under condition of using the normal distribution of feature changes with the standard deviation 0.1.

Figures 27 and 28 illustrate the heatmaps of the concordance index of the ER-SHAP-RF for the Breast Cancer dataset. They show results similar to the results obtained for the Boston Housing dataset demonstrated in Figures 20 and 21, respectively. This implies that the use of "pre-training" in the form of the random forest combined with the calibration method leads to a better approximation.

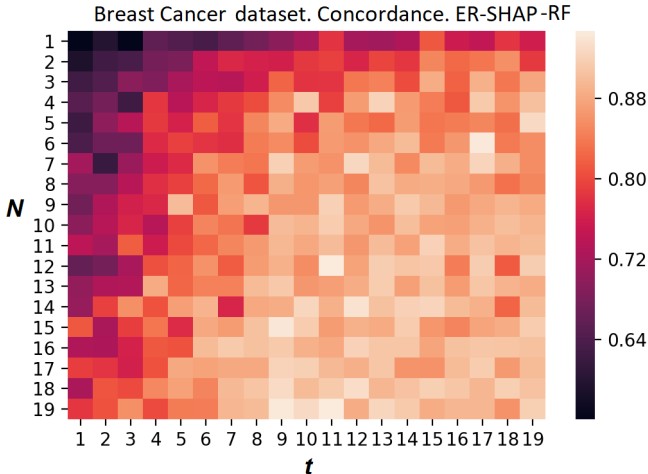

**Figure 27.** The heatmap reflecting the concordance index *C* obtained by ER-SHAP-RF for the Breast Cancer dataset without using the temperature scaling.

We also compare the proposed models with the SHAP and Kernel SHAP by using the following datasets. The California Housing dataset obtained from the StatLib repository consists of 20,640 instances such that each instance is described by eight features. It can be found in https://www.dcc.fc.up.pt/~ltorgo/Regression/cal_housing, accessed on 2 November 2022. The KDD Coil 7 dataset consists of 282 instances such that each instance is described by 11 features. The PBC dataset has 276 instances with 18 features. The Plasma Retinol dataset has 315 instances with 13 features. The Cholesterol dataset consists of

297 instances such that each instance is described by 13 features. The datasets KDD Coil 7, PBC, Plasma Retinol, and Cholesterol can be found at https://www.openml.org/search? type=data, accessed on 2 November 2022. We compare the proposed models with the Kernel SHAP and with the SHAP on these datasets by using the C-index. The number of the iteration $N$ and the number of the selected features $t$ are taken as 20 and 3, respectively, for all the datasets. The corresponding results are shown in Tables 1 and 2. It follows from Tables 1 and 2 that the ER-SHAP, ERW-SHAP, and ER-SHAP-RF provide almost the same results as the Kernel SHAP and SHAP because the values of the C-index are close to 1.

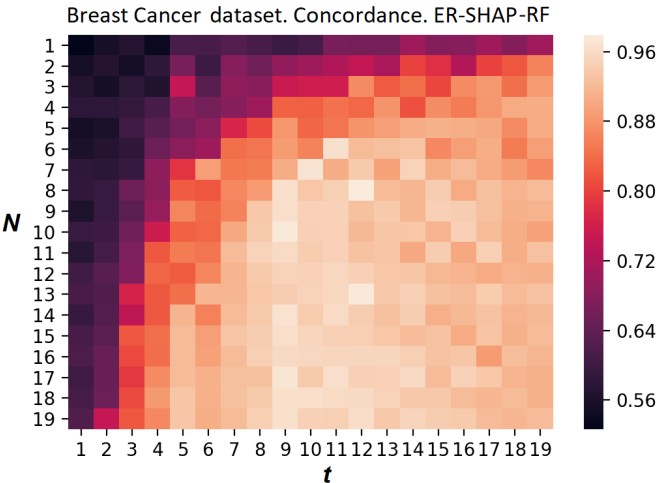

**Figure 28.** The heatmap reflecting the concordance index $C$ obtained by ER-SHAP-RF for the Breast Cancer dataset with using the temperature scaling.

**Table 1.** Comparison of the proposed models with Kernel SHAP for several datasets.

| Dataset | C-Index with Kernel SHAP | | |
|---|---|---|---|
| | **ER-SHAP** | **ERW-SHAP** | **ER-SHAP-RF** |
| California Housing | 0.990 | 0.996 | 1.000 |
| KDD Coil 7 | 0.876 | 0.980 | 0.982 |
| PBC | 0.926 | 0.908 | 0.948 |
| Plasma Retinol | 0.954 | 0.954 | 0.948 |
| Cholesterol | 0.926 | 0.944 | 0.974 |

**Table 2.** Comparison of the proposed models with SHAP for several datasets.

| Dataset | C-Index with SHAP | | |
|---|---|---|---|
| | **ER-SHAP** | **ERW-SHAP** | **ER-SHAP-RF** |
| California Housing | 0.974 | 0.984 | 0.994 |
| KDD Coil 7 | 0.872 | 0.978 | 0.978 |
| PBC | 0.908 | 0.886 | 0.934 |
| Plasma Retinol | 0.927 | 0.932 | 0.944 |
| Cholesterol | 0.922 | 0.938 | 0.970 |

## 6. Conclusions

It is important to note that only three modifications of the ensemble-based SHAP have been presented. At the same time, many additional modifications of the general approach

based on constructing the ensemble of SHAPs can be developed following the proposed modifications and the idea of the ensemble-based approximation.

First of all, the model of the feature selection used in the ER-SHAP-RF for "pre-training" can be changed. There are many methods solving the feature selection problem. Moreover, simple explanation methods can also be applied to the preliminary selection of the important features and to computing their probability distribution.

Second, various rules different from averaging can be applied to combining the results of the SHAPs, for example, the largest (smallest) Shapley values can be computed for providing pessimistic (optimistic) decisions.

The ensemble-based approach can be applied to an explanation of the classification as well as regression black-box models. It gives many opportunities for developing new methods which can be viewed as directions for further research. The proposed approach can be applied to local and global explanations. However, its main advantage is that it significantly reduces the computation time for solving the explanation problem.

**Author Contributions:** Conceptualization, L.U. and A.K.; methodology, L.U.; software, A.K.; validation, L.U. and A.K.; formal analysis, L.U.; investigation, A.K.; resources, L.U.; data curation, A.K.; writing—original draft preparation, L.U.; writing—review and editing, A.K.; visualization, A.K.; supervision, L.U.; project administration, L.U.; funding acquisition, L.U. All authors have read and agreed to the published version of the manuscript.

**Funding:** The research is partially funded by the Ministry of Science and Higher Education of the Russian Federation as part of the World-class Research Center program: Advanced Digital Technologies (contract No. 075-15-2020-934 dated 17 November 2020).

**Institutional Review Board Statement:** Not applicable.

**Informed Consent Statement:** Not applicable.

**Data Availability Statement:** Not applicable.

**Acknowledgments:** The authors would like to express their appreciation to the anonymous referees whose very valuable comments have improved the paper.

**Conflicts of Interest:** The authors declare no conflict of interest.

## Abbreviations

The following abbreviations are used in this manuscript:

| | |
|---|---|
| LIME | Local Interpretable Model-Agnostic Explanation |
| SHAP | SHapley Additive exPlanations |
| ER-SHAP | Ensemble of Random SHAPs |
| ERW-SHAP | Ensemble of Random Weighted SHAPs |
| ER-SHAP-RF | Ensemble of Random SHAPs generated by the Random Forest |
| SVM | Support Vector Machine |
| RBF | Radial Basis Function |
| C-index | Concordance index |

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
