# Peer review of "Ensembles of Random SHAPs"

_algorithms, doi:10.3390/a15110431_

Round 1

Reviewer 1 Report

* Section 2 seems to have only one subsection, which is strange. Maybe the subsection title should be removed, and the first paragraph rewritten to introduce the topic.

* The authors propose their method as very interesting in cases with a high number of features. But, when they use synthetic data, they use only 5 features. Why is that?

* I miss a direct comparison between the different ensemble versions that the authors propose, or summary of advantages/disadvantages of each of them after analyzing the results in the synthetic data.

* The authors use only 2 real-world datasets. In my opinion, it is a very low number. I would encourage authors to execute their proposals over a wider range of datasets, and provide the results (at least, using averaged results instead of providing them in such detail as the currently available datasets). This would increase the robustness of the results.

Reviewer 2 Report

The paper proposes an ensemble of explainability models of SHAP to reduce the number of features when dealing with a large number of features. This paper requires further improvement as follows.

(1) The presentation of the paper is a bit poor. For instance, there are many acronyms that have no proper explanation. Please provide a list of acronyms so that the reader may better understand.

(2) Figure 3 is unclear. Why the output of the black-box model becomes an input of a random forest? As a result, there is a loop here that is not clear to me. Isn't random forest generating the Shapley values too?

(3) Why do the authors consider synthetic data sets? I suggest using at least a real-world data set.

(4) In the Abstract, the authors mentioned that the original SHAP algorithm is computationally expensive. However, there is no comparison or computational cost analysis in the paper. I suggest the authors provide such a comparison.

(5) All the bar charts are too tiny so that hard to read.

(6) A comparison should be made with the existing XAI models, particularly how the proposed model outperforms other models.

Round 2

Reviewer 1 Report

The authors have addressed my previous comments, so I think that the paper is now ready to be published.

Reviewer 2 Report

The paper has been improved accordingly based on the previous reviewer's comment. The paper can be accepted after a minor check of the English.